# Optimal gestational weight gain and pregnancy outcomes, by BMI and height, in a marginalised population of women with short stature living along the Thailand-Myanmar border: A retrospective cohort, 2004–2023

Mary Gouws[1], Rose McGready[1,2]*, Wirichada Pan-ngum[1,3,4], Aung Myat Min[2], Nay Win Tun[2], Mary Ellen Gilder[1,2], Taco Jan Prins[5,6], Widi Yotyingaphiram[2], Mupawjay Pimanpanarak[2], Jacher Viladpai-Nguen[2], Nuttapol Panachuenwongsakul[2], François H. Nosten[1,2], Sue J. Lee[1,4]

1 Nuffield Department of Medicine, Centre for Tropical Medicine and Global Health, University of Oxford, Oxford, United Kingdom, 2 Shoklo Malaria Research Unit, Mahidol-Oxford Tropical Medicine Research Unit, Faculty of Tropical Medicine, Mahidol University, Mae Ramat, Thailand, 3 Department of Tropical Hygiene, Faculty of Tropical Medicine, Mahidol University, Bangkok, Thailand, 4 Mahidol-Oxford Tropical Medicine Research Unit, Faculty of Tropical Medicine, Mahidol University, Bangkok, Thailand, 5 Department of Family Medicine, Faculty of Medicine, Chiang Mai University, Chiang Mai, Thailand, 6 Department of Internal Medicine & Infectious Diseases, Amsterdam University Medical Centre, Research Groups: APH, GH and AII&I, Amsterdam UMC, Amsterdam, The Netherlands

* rose@shoklo-unit.com

## Abstract

### Background

Existing gestational weight gain (GWG) standards may not be applicable to women of short stature and those who have limited access to healthcare and are vulnerable to compromised nutrition during pregnancy. To inform the development of population-specific recommendations, this study investigated optimal GWG, by height and Body Mass Index (BMI), in a minority migrant and refugee population living along the Thailand-Myanmar border.

### Methods

Records of all women attending antenatal care in the first trimester at the Shoklo-Malaria Research Unit between 2004 and 2023 were retrospectively examined. GWG of 17,194 women was assessed against maternal, delivery and neonatal outcomes, by height and per Asia-Pacific BMI category. The Gestation Related Optimal Weight centiles were used to classify small, appropriate and large for gestational age neonates. Multivariable logistic regression analysis, including natural cubic splines, was used to assess the relationships between GWG and outcomes of interest. Optimal GWG per BMI group was defined as the GWG associated with the lowest composite

**Data availability statement:** The data cannot be shared publicly due to ethical restrictions: this data was routinely collected from a marginalized population of undocumented refugees and migrants and the women have not consented for the data to be shared. These restrictions are in keeping with the policy of the Oxford Tropical Research Ethics Committee. However, de-identified data is available from the Mahidol-Oxford Research Unit institutional data access committee upon reasonable request from researchers who meet the criteria for access to confidential data (contact Rita Chanviriyavuth, email rita@tropmedres.ac).

**Funding:** The Shoklo Malaria Research Unit is supported in part by the Wellcome-Trust Major Overseas Programme in Southeast Asia (# 220211, https://doi.org/10.35802/220211; lead applicant Nicholas Day). For the purpose of Open Access, the author has applied a CC BY public copyright licence to any Author Accepted Manuscript version arising from this submission. There was no additional external funding received for this study. The funders had no role in study design, data collection and analysis, decision to publish, or preparation of the manuscript.

**Competing interests:** The authors have declared that no competing interests exist.

risk for adverse outcomes. The optimal range included GWG values that did not exceed a 5% increase from the corresponding minimum composite risk.

## Results

Optimal GWG in women shorter than 153 cm was lower per BMI group than the National Academy of Medicine and Intergrowth-21 recommendations: underweight 12.1 kg (10.0–14.5), normal 10.4 kg (8.0–12.9), overweight/obese 5.3 kg (3.1–8.5); but comparable for women 153 cm or more: underweight 13.1 kg (11.0–15.1), normal 12.3 kg (9.7–15.3), overweight/obese 9.5 kg (6.4–13.4).

## Conclusion

Optimal GWG ranges are lower for this population with short stature compared to existing international guidelines. Clinical and contextual factors must be considered when implementing GWG recommendations for this, and other marginalized and short-stature populations.

## Introduction

Weight gain during pregnancy is widely recognized as an important determinant of maternal, delivery and neonatal outcomes [1–4]. Excessive gestational weight gain (GWG) has been linked to adverse outcomes such as gestational diabetes mellitus, hypertensive disorders of pregnancy, preterm birth, caesarean section and neonates born large for gestational age. Inadequate GWG has been associated with preterm birth and small for gestational age [3,4].

The National Academy of Medicine (NAM) formerly known as the Institute of Medicine (IOM) produced GWG recommendations in 1990, which were updated in 2009, and used data from women in the United States of America and Western Europe to define optimal GWG [5–7]. However, a 2016 review showed that the NAM recommendations were derived from a combination of studies, between which there was significant heterogeneity in methodology and statistical analysis [8].

To produce GWG guidelines across a diverse group of countries, data was used from the Intergrowth-21 (IG-21) prospective cohort [6]. This study incorporated 4,607 women and used strict inclusion and exclusion criteria to define a healthy pregnancy and neonatal outcome to create GWG guidelines for women with a normal BMI of 18.5–24.9 kg/m² (based on WHO BMI classification) and a height of 153 cm or more [6,9].

However, a 2021 study by Lee et al., in a marginalised group of women living along the Thailand-Myanmar found that short maternal stature was associated with lower GWG and suggested that height should be considered with first-trimester BMI when assessing ideal GWG during pregnancy [5]. This population has an average height of 151.5 cm, with almost 60% of women below the IG-21 cut-off for height of 153 cm [5]. Lee et al (2021) reported that appropriate for gestational age (AGA) in term newborns was achieved by 80% of women even though only 25% achieved GWG within NAM recommendations [5].

There is therefore a gap in the GWG literature for populations of short-stature who are at risk of compromised nutrition before and during pregnancy [5]. To inform the development of population-specific recommendations, this study investigated optimal GWG, by height and Body Mass Index (BMI), in a minority migrant and refugee population living along the Thailand-Myanmar border. The findings from our study may also be applicable to other populations of short stature globally, across socio-cultural contexts and healthcare services.

## Methods

### Ethical approval

Ethical approval was obtained through Oxford Tropical Research Ethics Committee (OxTREC reference 531−24) and the local body representing the community: Tak Province Community Advisory Board; T-CAB (TCAB202405). The T-CAB group is composed of community leaders who were consulted on the study design, process, and outcomes of interest [10].

### Study setting

Shoklo Malaria Research Unit (SMRU), based in Mae Ramat, Thailand, is a field-based research organisation [11]. SMRU undertakes a combination of humanitarian clinical services and context-relevant research on both sides of Thailand-Myanmar border [12]. SMRU offers free antenatal, labour and delivery care to marginalised populations on the border [12]. SMRU served both refugee and migrant populations, however, healthcare services for refugee women were handed over to another humanitarian organisation in 2016. Thus, SMRU now supports Thailand's Public Health Department by filling a gap in care for migrant and displaced populations.

### Study design

All pregnancy records from women with singleton pregnancies born at 28 weeks or later, between 01/01/2004 and 31/12/2023 were extracted from the SMRU electronic database. Antenatal and birth records are routinely collected and entered in a central electronic database which is securely managed by a local data team. The clinical information was gathered from medical documents that were originally maintained on paper and transitioned to electronic format in late 2008. Records on paper between 1986 and 2008 were digitized and incorporated into an electronic database to facilitate retrieval of antenatal care and birth outcome data. After 2008, all records have been entered directly into the electronic database.

The dataset for the purposes of this research was first accessed on 23/06/2024 and data extraction from the database was completed by 30/06/2024. When required for clarification, paper-based records were checked, which are stored at the SMRU offices. This was completed by 30/07/2024. All records were anonymised and identifiable information from the women was removed prior to analysis.

### Participant inclusion/exclusion and study variables

A process of data extraction based on inclusion and exclusion criteria was followed. Only records of singleton pregnancies with an estimated gestational age (EGA) of less than 14 weeks at first antenatal visit, and at least two weight measurements over the course of the pregnancy were included. Women with last maternal weight recorded >4 weeks prior to delivery, pre-existing conditions such as hypertension, diabetes mellitus, HIV, moderate to severe anaemia at the first antenatal care visit and delivery prior to 28 weeks' gestation (spontaneous abortion) were excluded. Iatrogenic preterm births and neonates with major congenital abnormalities were also excluded. The small number of neonates with major congenital abnormalities were excluded due to their possible skewed birth weight distributions, such as being small or large for gestational age, which could confound the assessment of the relationship between maternal weight gain and neonatal outcomes. Additionally, certain congenital abnormalities can directly or indirectly influence GWG.

First trimester BMI (<14 weeks gestation) was used to represent pre-pregnancy BMI, as it has been shown to be an accurate approximation in several international cohorts [13–15]. The date of measurement of last maternal weight was used to calculate GWG and was required to be within four weeks of delivery to be included in the analysis. Trained mid-wives collected weight measurements at first ANC consultation and at each follow-up visit using mechanical Salter scales with 0.5kg precision. Gestational age was calculated by ultrasound, using crown-rump length between 9 + 0 to 13 + 6 weeks. The quality of scanning in this population has been confirmed and is reliable [16,17]. Similar to the IG-21study, short stature was defined as maternal height less than 153cm [6,18].

SGA, AGA and LGA classification was undertaken by calculating sex-, ethnicity- and gestational age-adjusted birth-weight centiles using the Perinatal Institute's Gestation Related Optimal Weight (GROW) customised bulk centile calculator V.8.0 [19]. GROW offers the advantage of region-specific classification and has coefficients to represent over 100 country-of-origin groups. Based on the average height, the Myanmar standards were used for women of Burmese ethnicity, and the Nepalese standards most closely matched the average height of Karen and other ethnicities in the dataset [20].

Neonates below the 10th percentile were classified as SGA, while those above the 90th percentile were classified as LGA [21,22]. Neonates at and between the 10th and 90th percentiles were classified as AGA.
Based on the literature and the availability of data in the cohort, the following maternal, birth and neonatal outcomes were considered for further analysis in relation to GWG: gestational diabetes mellitus (GDM), hypertensive disorders of pregnancy (HDoP), caesarean section (CS), preterm birth (birth <37 weeks' gestation), small-for gestational age (SGA) and large-for gestational age (LGA). HDoP included gestational hypertension, pre-eclampsia and eclampsia.

## Statistical analysis

All statistical analyses were conducted using R (Version 4.2.1) and RStudio (Version 2024.04.0 + 735). Initial descriptive statistics were performed according to Asia-Pacific BMI categories which were defined as: underweight (<18.5 kg/m²), normal weight (18.5–22.9 kg/m²), overweight (23.0–24.9 kg/m²), and obese (≥25.0 kg/m²) [23,24]. For the purposes of this analysis, overweight and obese women were categorized into a single group as the narrow overweight range limited its utility as a distinct category and separating the overweight and obese groups resulted in small, underpowered groups.

Continuous variables with a normal distribution were summarised using mean ± standard deviation (SD), otherwise median and interquartile range (IQR; 25th-75th percentiles) were reported. Categorical variables were expressed as frequencies with percentages (n(%)). The chi-squared test and Cochran-Armitage test for trend were used to assess associations between categorical variables across BMI groups. Statistical significance was set at $p < 0.05$.

Univariable and multivariable logistic regression models were used to assess the relationships between GWG and outcomes. Multivariable models were adjusted for known confounders including age, parity, smoking status and ethnicity, based on the literature [3,25]. To address potential collinearity between age and parity, a categorisation to combine age and parity into six groups was created, and was used as a single categorical variable in the models. The groups were defined as age < 20 and nulliparous; age < 20 and parity ≥1; age 20–29 and nulliparous; age 20–29 and parity ≥1; age ≥ 30 and nulliparous; age ≥ 30 and parity ≥1.

Adjustment was also made for year (2004–2012, 2013–2016, 2017–2020, 2021–2023). This accounted for method of GDM screening, as this changed from no screening before 2013, to risk-factor based screening in 2013–2020, to full population screening after 2020 [26] and also for the change in care provision to migrant populations only from 2017. Additionally, adjustment was made for malaria (treated during pregnancy), as it is a common infection in this area, and is associated with adverse outcomes including HDoP, SGA, preterm birth and stillbirth [16,27,28]. Although literacy and socioeconomic status are also possible confounders [4,25], reliable measures of these variables were not available within the dataset.

To allow for potential nonlinear relationships between GWG and adverse outcomes, logistic regression models were fitted incorporating natural (restricted) cubic splines for GWG. Natural cubic splines were used to mitigate overfitting, especially at the boundaries of the data [29]. The splines were constructed as a series of polynomial functions, each defined within a specific interval of GWG. Each spline was joined smoothly using knots, ensuring that the overall function was continuous. The optimal number of knots was determined though visualisation of the relationship between predictor and outcome variables. All models were constructed using 2 to 3 degrees of freedom (i.e., 3 to 4 knots) to maintain simplicity while capturing essential patterns.

Using these models, predicted probabilities were calculated for each adverse outcome across the range of GWG. For each weight, the predicted probabilities of all adverse outcomes were summed to derive a composite risk score. The minimum composite risk score was used to define the optimal GWG, with the optimal range corresponding to a composite score within 5% of the minimum risk. This methodology uses elements similar to those used by Choi et al. (2017) for a cohort of Korean women, by Morisaki et al. (2017) who considered GWG in Japanese women, and by Ee et al., in a Singaporean cohort (2014) [30–32].

## Results

Fig 1 illustrates the selection process for the study cohort. From a total of 44,364 women with singleton pregnancies and deliveries at ≥28 weeks' gestation between January 1, 2004, and December 31, 2023, 25,565 women (58.0%) were excluded due to enrolment at ≥14 weeks' gestation (n = 24,016) or incomplete height and weight data (n = 1,549). Additionally, women with last maternal weight recorded >4 weeks prior to delivery (n = 1,262), pre-existing conditions such as hypertension, diabetes mellitus, or HIV (n = 75), moderate to severe anaemia at the first antenatal care visit (HCT < 25%, n = 48), iatrogenic preterm births (N = 174) and neonates with major congenital abnormalities (n = 46) were excluded. The final cohort analysed consisted of 17,194 women.

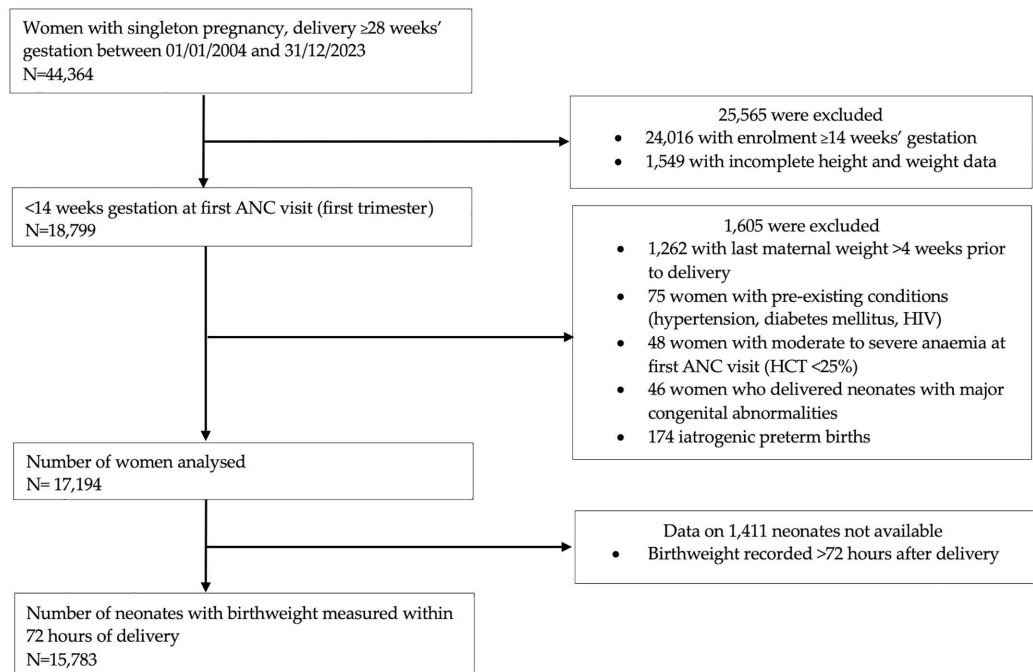

**Fig 1. Flow of study participants.** Key: ANC = antenatal care; HIV = Human Immunodeficiency Virus; HCT = haematocrit.

## Maternal demographic characteristics

Table 1 shows the characteristics, by Asia-Pacific BMI group, of the 17,194 women who met the inclusion criteria for the study between 2004 and 2023. More than half of women, 10,115 (58.8%) had a normal first trimester BMI, while 3,019 (17.6%) were underweight (minimum BMI 13.5 kg/m²), and 4,060 (23.6%) overweight or obese by Asia-Pacific BMI standards (Table 1). There were 2,089 women with an overweight BMI (23–24.9 kg/m²) and 1,971 women with an obese (≥ 25 kg/m²) first trimester BMI (S1 Table in S1 File).

In the underweight BMI group, the range was 4.9 kg/m² (13.5 to 18.4 kg/m²) and the range for the normal BMI category was bound at 4.4 kg/m² (18.5–22.9 kg/m²). In the overweight BMI group, the range was 1.9 kg/m² (23–24.9 kg/m²), while for the obese group it was much wider, 18 kg/m² (25–43 kg/m²). The proportion of women in the cohort with BMI ≥ 30 kg/m² (obese by WHO International Standards) was only 2% (356 of 17,194 women). The distribution of BMI can be seen in S1 Fig in S1 File.

The mean height (SD) for the cohort was 151.4 cm (5.4 cm), and more than half of women (10,205, 59.4%) had a height less than 153 cm. The distribution of height in the cohort can be seen in the supporting information document (S2 Fig in S1 File). Mean GWG for the full cohort was 9.3 kg (SD 4.0 kg), which decreased with increasing BMI category (test for trend p < 0.001). Most women with a BMI < 18.5 kg/m² gained below NAM recommendations 2,266 (75.1%); only a quarter 696 (23.1%) gained within NAM guidelines. A similar pattern was present in the women with normal BMI, where 7,305 (72.2%) gained below NAM guidelines, while 2,363 (23.4%) gained within NAM recommendations, and 447 (4.4%) gained above NAM recommendations. In the overweight and obese group almost half 1,755 (43.2%) of women gained within NAM recommendations, with 1,154 (28.4%) gaining below and 1,151 (28.3%) gaining above NAM recommendations.

**Table 1. Maternal demographic and outcome descriptive statistics by Asia-Pacific BMI categories.**

| Variable | All Women, n (%) | Underweight (<18.5 kg/m²) | Normal (18.5–22.9 kg/m²) | Overweight & Obese (≥23 kg/m²) |
|---|---|---|---|---|
| **Total Women** | 17,194 | 3,019 (17.6) | 10,115 (58.8) | 4,060 (23.7) |
| **Median Age (IQR), years** | 25 (20–30) | 23 (20–28) | 24 (20–30) | 28 (23–33) |
| **Nulliparous (%)** | 6231 (36.2) | 1345 (44.6) | 3933 (38.9) | 953 (23.5) |
| **Ethnicity** | | | | |
| Karen (%) | 8,994 (52.3) | 1,383 (45.8) | 5,427 (53.7) | 2,184 (53.8) |
| Burmese (%) | 4,160 (24.2) | 884 (29.3) | 2,154 (21.3) | 1,122 (27.6) |
| Other (%) | 4,040 (23.5) | 752 (24.9) | 2,534 (25.1) | 754 (18.6) |
| **Mean Height (SD), cm** | 151.4 (5.4) | 152 (5.5) | 151.1 (5.4) | 151.8 (5.4) |
| **Height <153 cm (%)** | 10,205 (59.4) | 1,692 (56.0) | 6,198 (61.3) | 2,315 (57.0) |
| **Mean GWG (SD), kg** | 9.3 (4.0) | 10.2 (3.5) | 9.5 (3.8) | 8.2 (4.4) |
| **NAM GWG Recommendation (kg)** | | | | |
| Below NAM (%) | 10,725 (62.4) | 2,266 (75.1) | 7,305 (72.2) | 1,154 (28.4) |
| Within NAM (%) | 4,814 (28.0) | 696 (23.1) | 2,363 (23.4) | 1,755 (43.2) |
| Above NAM (%) | 1,655 (9.6) | 57 (1.9) | 447 (4.4) | 1,151 (28.3) |
| **Smokers (%)[a]** | 2,650 (15.4) | 588 (18.5) | 1,673 (16.5) | 419 (10.3) |
| **Hypertensive Disorder (%)** | 1,127 (6.6) | 110 (3.6) | 557 (5.5) | 460 (11.3) |
| **Gestational Diabetes Mellitus (%)[b]** | 628 (3.7) | 77 (2.6) | 219 (2.2) | 332 (8.2) |
| **Malaria in Pregnancy (%)** | 1,703 (9.9) | 385 (12.8) | 1,077 (10.6) | 241 (5.9) |
| **Caesarean Section (%)** | 857 (5.0) | 86 (2.8) | 405 (4.0) | 366 (9.0) |

[a]Smoking, n=47 values missing. [b]Gestational Diabetes Mellitus, n=7 values missing.

BMI = Body Mass Index; NAM = National Academy of Medicine; GWG = gestational weight gain; IQR = interquartile range; SD = standard deviation; kg = kilograms; cm = centimetres; m = metres.

Most women had a vaginal birth 16,337 (95.0%) while 857 (5.0%) delivered by caesarean section. Rates of caesarean section increased with increasing BMI category from 2.8% (underweight group) to 9.0% (overweight/obese group) (test for trend p<0.001).

### Delivery and neonatal outcomes

Birth outcome was available for all 17,194 neonates (Fig 1), of which 17,109 (99.5%) were live births (Table 2). Of all neonates, 1,048 (6.1%) were born preterm. Birthweight for gestational age centiles were calculated for 15,783 (91.8%) neonates who had a birthweight measured within 72 hours of delivery. While most babies were born AGA 12,402 (78.6%), nearly one in six were SGA 2,463 (15.6%), and a minority, one in twenty 918 (5.8%) were LGA. S2 Table in S1 File presents neonatal outcomes by BMI category separated for overweight and obese categories.

### Calculation of optimal GWG

Two thirds of the pregnancies did not experience HDoP, GDM, CS, spontaneous preterm birth, SGA or LGA 11,309 women (65.8%), 4,878 (28.4%) had one adverse outcome, and less than 6% experienced two or more adverse outcomes 1,007 women (5.6%). The six outcomes of HDoP, GDM, CS, spontaneous preterm birth, SGA and LGA were statistically significantly associated with GWG, P<0.001 (S3 Table in S1 File). All six outcomes were plotted as predicted probabilities against GWG using smoothed restricted cubic splines for the BMI categories of underweight, normal and overweight/obese, each for two height categories of <153 and ≥ 153cm (Figs 2-4). The GWG at which the composite risk score for adverse outcomes was lowest was used to calculate an optimal GWG, and range (Table 3).

In the cohort of women with an underweight BMI in the first trimester and height <153cm, the optimal GWG was 12.1 (10.0-14.5kg), while in the underweight group with height of 153cm or more, the optimal GWG was higher, at 13.1kg (11.0-15.1kg) (Fig 2). A similar pattern was seen across both the normal BMI and overweight/obese groups, in which the women with a height <153cm had a lower optimal GWG than the taller women (Figs 3–4). In the women with a normal first trimester BMI, optimal GWG in the <153cm cohort was 10.4kg (8.0-12.9kg), while for normal BMI women with height ≥153cm, optimal GWG was 12.3kg (9.7-15.3kg) (Fig 3). In the combined overweight/obese

**Table 2. Neonatal descriptive statistics by maternal Asia-Pacific BMI categories.**

| Variable | All | Underweight (<18.5 kg/m²) | Normal (18.5–22.9 kg/m²) | Overweight & Obese (≥23 kg/m²) |
|---|---|---|---|---|
| **All births with known outcome (%)** | 17,194 | 3,019 (17.6) | 10,115 (58.8) | 4,060 (23.6) |
| Live births | 17,109 (99.5) | 2.999 (99.3) | 10,065 (99.5) | 4,045 (99.6) |
| Stillbirths | 85 (0.5) | 20 (0.7) | 50 (0.5) | 15 (0.4) |
| Male[a] | 8,648 (50.3) | 1,494 (49.5) | 5,096 (50.4) | 2,058 (50.7) |
| Median EGA (IQR), weeks | 39.3 (38.4–40.1) | 39.2 (38.3–40.0) | 39.3 (38.4–40.1) | 39.4 (38.5–40.2) |
| Preterm births <37 weeks (%) | 1,048 (6.1) | 259 (8.6) | 645 (6.4) | 144 (3.5) |
| **Birthweight measured <72 hours[b]** | 15,783 | 2,672 (16.9) | 9,284 (58.8) | 3,827 (24.2) |
| Mean Birthweight (SD), g | 2,982.0 (446.8) | 2,843.5 (440.9) | 2,959.0 (434.4) | 3,134.6 (438.3) |
| SGA (%) | 2,463 (15.6) | 470 (17.6) | 1,414 (15.2) | 579 (15.1) |
| AGA (%) | 12,402 (78.6) | 2,061 (77.1) | 7,333 (79.0) | 3,008 (78.6) |
| LGA (%) | 918 (5.8) | 141 (5.3) | 537 (5.8) | 240 (6.3) |

[a]Male baby, n=6 values missing.

[b]Birthweight for gestational age centiles were calculated for 15,783 (91.8%) neonates who had a birthweight measured within 72 hours of delivery.

BMI=Body Mass Index; EGA=estimated gestational age; SGA=small for gestational age; AGA=appropriate for gestational age; LGA=large for gestational age; IQR=interquartile range; SD=standard deviation; kg=kilograms; g=grams; m=metres.

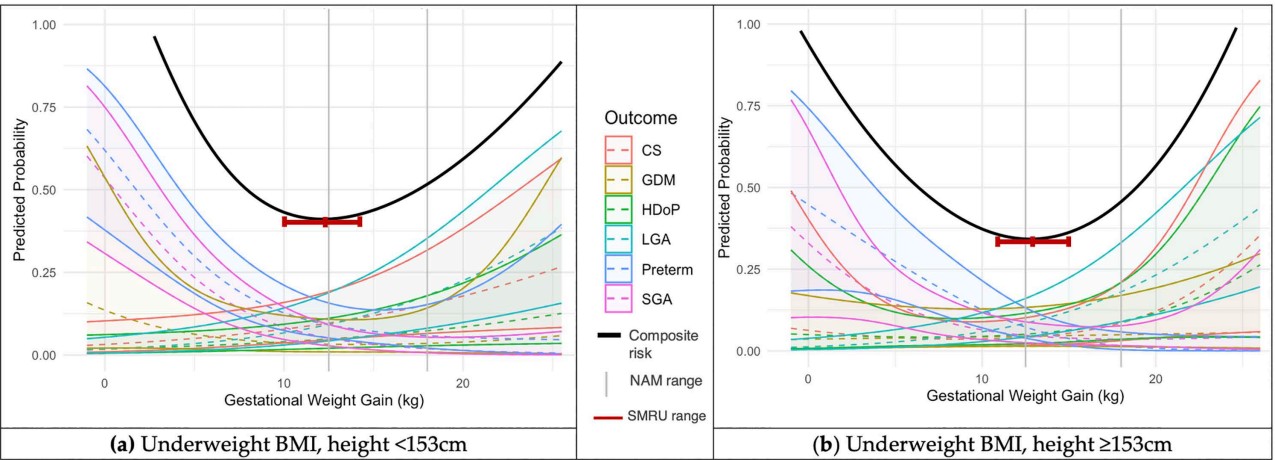

**Fig 2. Predicted probabilities with 95% confidence intervals (y-axis) of adverse outcomes across GWG (x-axis) for women with an underweight first trimester BMI.** Calculated using logistic regression models incorporating cubic splines. NAM GWG recommendations superimposed as vertical grey lines at 12.5 and 18.0 kg (a) Women with underweight BMI and height <153 cm, N = 1,692; (b) Women with underweight BMI and height ≥153 cm, N = 1,327. Key: NAM = National Academy of Medicine; BMI = Body Mass Index; CS = Caesarean Section; GDM = Gestational Diabetes Mellitus; HDoP = Hypertensive Disorders of Pregnancy; LGA = Large for Gestational Age; Preterm = Preterm birth; SGA = Small for Gestational Age; cm = centimetres; kg = kilograms.

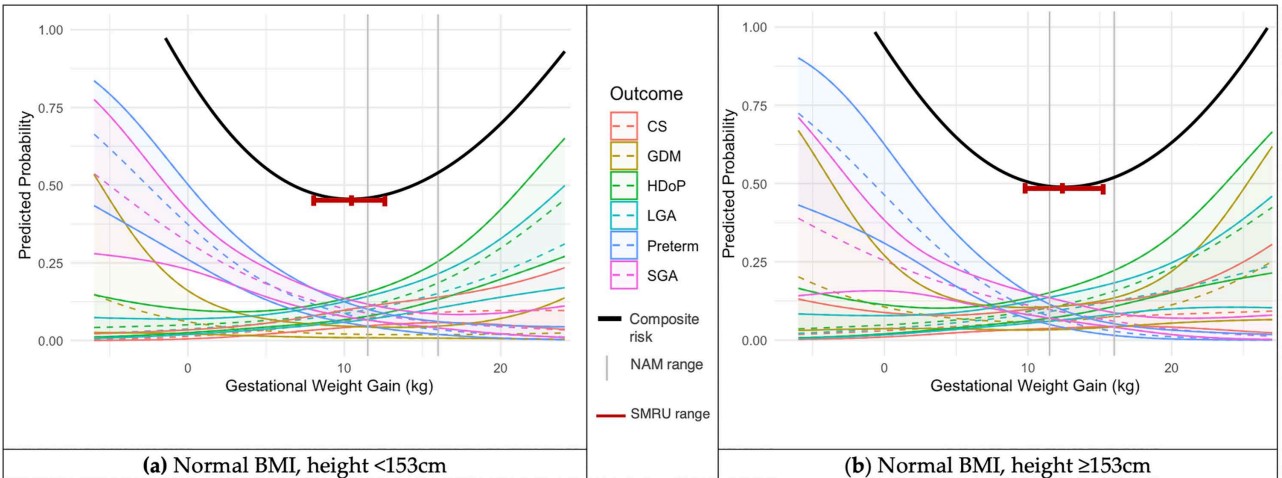

**Fig 3. Predicted probabilities with 95% confidence intervals (y) of adverse outcomes across GWG (x) for women with a normal first trimester BMI.** Calculated using logistic regression models incorporating cubic splines. NAM GWG recommendations superimposed as vertical grey lines at 11.5 and 16.0 kg (a) Women with normal BMI and height <153 cm, N = 6,198; (b) Women with normal BMI and height ≥153 cm, N = 3,917. Key: NAM = National Academy of Medicine; BMI = Body Mass Index; CS = Caesarean Section; GDM = Gestational Diabetes Mellitus; HDoP = Hypertensive Disorders of Pregnancy; LGA = Large for Gestational Age; Preterm = Preterm birth; SGA = Small for Gestational Age; cm = centimetres; kg = kilograms.

first-trimester BMI group, in the women with height <153cm, optimal GWG was 5.3kg (3.1-8.5kg), while in those ≥153cm, optimal GWG was 9.5kg (6.4-13.4kg) (Fig 4). S3 Fig, S4 Fig and S5 Fig in S1 File show the calculated optimal GWG ranges superimposed on the predicted probabilities of adverse maternal and neonatal outcomes, stratified by BMI and height categories.

**Table 3. Calculated optimal GWG ranges based on minimisation of composite risk of adverse outcomes, per Asia-Pacific BMI category, by height (<153 cm vs. ≥153 cm) and compared with NAM GWG recommendations.**

| Asia-Pacific BMI category (kg/m²) | Optimal GWG (range), kg full cohort (n=17,194) | Optimal GWG (range), kg height <153 cm (n=10,205) | Optimal GWG (range), kg height ≥153 cm (n=6989) | NAM GWG recommendation, kg[a] |
|---|---|---|---|---|
| Underweight (<18.5) | 12.4 | 12.1 | 13.1 | 12.5-18.0 |
| | (10.2–14.8) | (10.0-14.5) | (11.0-15.1) | |
| | N=3,019 | N=1,692 | N=1,327 | |
| Normal (18.5–22.9) | 11.0 | 10.4 | 12.3 | 11.5-16.0 |
| | (8.3–14.0) | (8.0-12.9) | (9.7-15.3) | |
| | N=10,115 | N=6,198 | N=3,917 | |
| Overweight/ Obese (≥23) | 9.0 | 5.3 | 9.5 | Overweight: 7–11.5 Obese: 5.0–9.0 |
| | (5.6-12.1) | (3.1-8.5) | (6.4-13.4) | |
| | N=4,060 | N=2,315 | N=1,745 | |

[a]For reference: NAM GWG recommendations are based on WHO BMI classifications which are defined as follows: underweight (<18.5 kg/m²), normal (18.5–24.9 kg/m²), overweight (25.0–29.9 kg/m²), obese (≥30.0 kg/m²).

GWG = gestational weight gain; BMI = Body Mass Index; NAM = National Academy of Medicine; WHO = World Health Organization; kg = kilograms; cm = centimetres; m = metres.

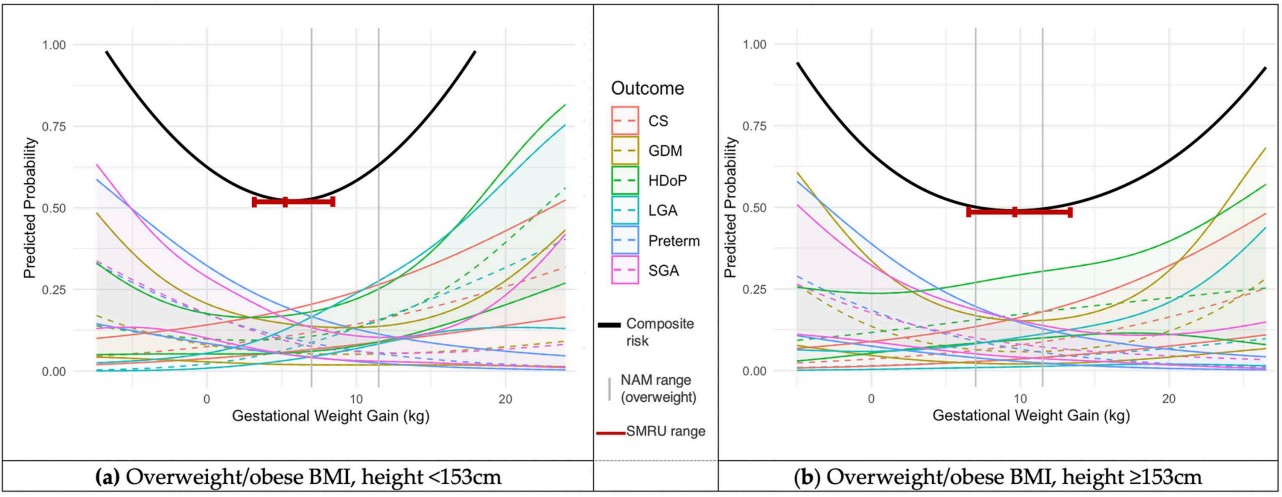

**Fig 4. Predicted probabilities with 95% confidence intervals (y) of adverse outcomes across GWG (x) for women with an overweight or obese first trimester BMI.** Calculated using logistic regression models incorporating cubic splines. NAM GWG recommendations superimposed as vertical grey lines at 7.0 and 11.5 kg (a) Women with overweight or obese BMI and height <153 cm, N=2,315; (b) Women with overweight or obese BMI and height ≥153 cm, N=1,745. Key: NAM = National Academy of Medicine; BMI = Body Mass Index; CS = Caesarean Section; GDM = Gestational Diabetes Mellitus; HDoP = Hypertensive Disorders of Pregnancy; LGA = Large for Gestational Age; Preterm = Preterm birth; SGA = Small for Gestational Age; cm = centimetres; kg = kilograms.

## Discussion

GWG ranges based on first-trimester Asia-Pacific BMI categories were calculated to minimize the risks of adverse pregnancy and birth outcomes. In general, the GWG ranges for the migrant and refugee population were lower than the NAM guidelines. For women with height less than 153cm and normal BMI, the optimal GWG was 10.4kg with range 8.0-12.9 kg compared with 11.5-16.0 from the NAM guidelines.

The NAM guidelines, primarily derived from women in North America (average height of 162 cm) and Western Europe (166 cm), recommend a GWG of 11.5-16kg at term in normal BMI women by WHO BMI standards [7,20]. IG-21, which included only women with a height of 153cm or more, provide a 50th centile GWG recommendation of 13.2kg (10th-90th centiles: 8.3-19.4kg) for normal BMI women (by WHO BMI standards) at 39 weeks' gestation (equivalent to the mean gestation at delivery in the migrant and refugee cohort) [6]. In this analysis, when restricted to women of 153cm or more, optimal GWG (12.3kg, range: 9.7-15.3) was comparable to the IG-21 range. However, for women shorter than 153cm, the optimal GWG was lower (10.4kg, range: 8.0-12.9). For the underweight BMI category, in which NAM recommends a GWG of 12.5-18kg, in this analysis optimal GWG was 13.1kg (11.0-15.1) in women ≥153cm, while in underweight women <153cm, optimal GWG was lower: 12.1kg (10.0-14.5). One reason for this difference may be because while this study considered the outcomes of HDoP, GDM, CS, spontaneous preterm birth, SGA and LGA, the NAM GWG recommendations additionally considered postpartum weight retention and childhood obesity, but not GDM [7]. Data on postpartum weight retention and childhood obesity were not available for analysis in this study.

Morisaki et al. (2017) included 104,070 Japanese women with singleton pregnancies to find the GWG with lowest risk of adverse outcomes (SGA, preterm birth, pre-eclampsia and complicated delivery: CS, forceps or vacuum delivery, obstructed labour, PPH) [31]. Morisaki et al. calculated the optimal weight gain to minimise adverse outcomes, by BMI groups (kg/m²): BMI 17.0–18.4: 12.2kg; BMI 18.5–19.9: 10.9kg; BMI 20–22.9: 9.9kg; 23–24.9: 7.7kg; and 25–27.4: 4.3kg [31]. This Japanese population had a mean height of 158cm (lower than the average for American and European populations) and the optimal GWG ranges were close to those developed for the cohort of women with short stature analysed here [31].

An additional finding in the current analysis of women living along the Thailand-Myanmar border is that the proportion of women with BMI ≥30 kg/m² is only 2%, which contrasts with countries such as the US, where this proportion is estimated to be higher than 20% [33,34]. In the Thai-Myanmar border population, the overall GWG distribution may be skewed by the higher proportion of women at the lower end of the BMI spectrum. This finding highlights the role that BMI distribution plays in shaping GWG patterns within a population and may also explain why the interval for GWG in the women with first trimester overweight and obese BMI was wider than that calculated for normal and underweight BMI groups (Table 3). This wider interval can be attributed to the wide variation of first trimester BMI in the overweight and obese group, which ranged from 23 to 43kg/m². In the underweight BMI group, the range was only 4.9kg/m² (13.5 to 18.4 kg/m²) and the range for the normal BMI category was bound at 4.4 kg/m² (18.5-22.9 kg/m²). Thus, applying guidelines developed in populations with different BMI distributions may not fully capture the nuanced needs and unique demographic and health characteristics of all populations.

The relationship between maternal height, birth outcomes and BMI is complex. The women in this study represent a marginalised population where nutrition may be compromised but there is also an increasing proportion of women with overweight or obese BMIs [35]. This "double burden of malnutrition" [36]combined with short stature may impact the outcomes considered in this study, including caesarean section [37,38], small for gestational age and large for gestational age [39]. Indeed, Rahman et al reported that maternal overweight combined with short stature was associated with a higher risk of caesarean section regardless of socio-economic status which usually predicts risk for this outcome [37]. Yearwood et al found maternal height modified the association between SGA and LGA and risk of adverse outcomes; increased maternal height was associated with increased adverse outcomes in SGA babies but decreased risk in LGA babies [39]. Although we know that all women in this cohort were seeking services in a humanitarian healthcare setting, data on literacy and socioeconomic status, which can influence health and well-being of pregnant women and affect the GWG, was limited for this cohort, thus could not be adjusted for in the analysis. We were also unable to adjust for within subject correlation due to repeat pregnancies (i.e., mothers being included in the dataset more than once). Data collection for this was only available for more recent pregnancies and as data collection continues, this is an aspect of analysis that will be included in future studies.

An additional limitation to the approach used for calculating optimal GWG in this analysis is that the composite scores are likely overestimating risk because the interrelated nature of certain adverse outcomes is not accounted for. For instance, GDM, CS and LGA may co-occur and the calculation of the composite score which considers each of these as separate risks might overestimate the total calculated risk. However, in this cohort the proportion of women with two or more adverse outcomes was less than 6%, thus, the impact of "overestimating" risk was likely minor. Instead, the GWG ranges reported here would be most strongly influenced by the two thirds of women who had no adverse events.

Additionally, the screening and diagnosis of GDM has changed over time in this population, and thus the estimation of risk for GDM may not be representative of the true rates in this population. Future analyses should consider an approach to account for the potential overlap between adverse outcomes to provide a more accurate risk assessment.

To get an accurate reflection of GWG patterns and classify women into BMI groups, the data in this analysis was limited to women who had their first antenatal care (ANC) visit before 14 weeks' gestation. This process excluded approximately 40% of women who gave birth between 2004 and 2023, reflective of the real-world situation where many women do not access care early in gestation or do not have a documented first trimester BMI. If, for example, a woman at 20 weeks' estimated gestational age presents for a first time ANC visit, already having gained weight, it would be unclear which recommendations should be followed. Additionally, classifying BMI, a continuous variable, into groups comes at the expense of classifying women with a BMI of 22.8 and 23.0 as more different than similar, and then applying differing GWG recommendations to these women. Likewise, the height categories applied in this analysis were broad and may lead to recommendations that do not apply well to rare outliers (<140 cm or >160 cm) in this population.

A third challenge of BMI categorisation was the sample sizes per BMI group. The relatively smaller number of women with overweight (N=2,089) and obese (N=1,971) first-trimester BMIs meant that when the analysis was run for these groups individually the results were underpowered. Hence overweight and obese were combined for the analysis although overweight and obese women are not a homogenous group and ideally these groups would be analysed separately.

To address the limitations of BMI categorisation, further research should focus on the development of an interactive tool into which height, current weight and gestation can be entered, with an output of optimal GWG based on these criteria. Alternatively, methods of estimating GWG which do not rely on first trimester or pre-pregnancy BMI could also be considered. One example of this is mid-upper arm circumference (MUAC) measurement, which was proven to be strongly associated with first-trimester BMI and GWG in Darling et al.'s 2023 systematic review and meta-analysis [25]. Future studies could consider how MUAC changes over gestation and if MUAC can be used to accurately represent BMI or GWG patterns, especially in women who do not enter antenatal care in first trimester.

## Conclusion

This study contributes to the growing body of literature on GWG by focusing on a unique and marginalized population of women of short-stature living along the Thailand-Myanmar border. The international NAM guidelines, based on taller populations from North America and Western Europe, overestimate the optimal GWG for this migrant and refugee cohort, which has an average height of 151.4cm. The GWG ranges developed in this study can be applied in clinical practice when advising women attending antenatal care at SMRU on the amount of weight to gain during pregnancy. There are limited dataon pregnant women with short stature globally, thus the findings from our study may also have applicability to other populations of short stature worldwide.

## Supporting information

**S1 File.  S1-S5 Figs, S1-S3 Table.**
(PDF)

## Acknowledgments

We recognize the efforts of the maternal and child health team at SMRU whose commitment to patient care and data collection has been foundational to this research and represents the collective work of many people over many years.

## Author contributions

**Conceptualization:** Mary Gouws, Rose McGready, Wirichada Pan-ngum, Sue J Lee.

**Data curation:** Rose McGready, Aung Myat Min, Nay Win Tun, Mary Ellen Gilder, Taco Jan Prins, Widi Yotyingaphiram, Mupawjay Pimanpanarak, Jacher Viladpai-Nguen, Nuttapol Panachuenwongsakul, François H Nosten.

**Formal analysis:** Mary Gouws, Sue J Lee.

**Methodology:** Mary Gouws, Sue J Lee.

**Supervision:** Rose McGready, Wirichada Pan-ngum, Sue J Lee.

**Writing – original draft:** Mary Gouws.

**Writing – review & editing:** Mary Gouws, Rose McGready, Wirichada Pan-ngum, Aung Myat Min, Nay Win Tun, Mary Ellen Gilder, Taco Jan Prins, Widi Yotyingaphiram, Mupawjay Pimanpanarak, Jacher Viladpai-Nguen, Nuttapol Panachuenwongsakul, François H Nosten, Sue J Lee.

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
